# Reverse Design of Solid Propellant Grain for a Performance-Matching Goal: Shape Optimization via Evolutionary Neural Network

Wentao Li [1], Wenbo Li [2], Yunqin He [1] and Guozhu Liang [1],*

1   School of Astronautics, Beihang University, Beijing 102206, China
2   School of Aerospace Engineering, Tsinghua University, Beijing 100084, China
*   Correspondence: lgz@buaa.edu.cn; Tel.: +86-010-61716982

**Abstract:** The reverse design of solid propellant grain for a performance-matching goal, one of the most challenging directions of the solid rocket motor designing work, is limited by the traditional semi-empirical parameter-driven optimization methods based on some predefined grain configurations. Grain designers call for a new method that can automatically provide brand-new grain shapes beyond the traditional ones. In this work, a shape optimization method based on the evolutionary neural network is proposed to achieve the reverse design of two-dimensional (2D) grains. Firstly, the modified ellipse-form eikonal equation is solved by using the finite element method to realize the burn-back analysis of 2D grains in any shape on a fixed unstructured mesh. Then, the neural network is introduced to determine the spatial distribution of the propellant to define the grain shape. The hyperparameters of the network are continuously evolved with the aid of the genetic algorithm. Finally, the optimal grain shape that matches the performance goal most is obtained. The method is verified in different scenarios. The result shows that the design can precisely match the given pressure-time curve of star grains and slotted-tube grains. Furthermore, the method can automatically evolve a new dendritic-shaped grain that matches the given dual-thrust pressure-time curve. Since the reverse design uses the concept of shape optimization, it does not require any pre-selection of the grain shape, and the designers shall be free from defining different kinds of geometric parameters for specific grain configurations. Consequently, the method has the potential to apply in the reconstruction of an actual grain and the conceptual design of innovative grain configurations.

**Keywords:** solid rocket motor; grain design; reverse design; shape optimization; evolutionary neural network; grain burn-back analysis; finite element method; genetic algorithm



## 1. Introduction

Propellant grain design is a significant stage of solid rocket motor designing work and its major task is to match the given performance curves. The main internal ballistic performance curves include the variations of combustion chamber pressure with time ($p_c{\sim}t$ curve) and motor thrust with time ($F{\sim}t$ curve). These curves are essential to the overall design of the missile/rocket and even the guidance, navigation, and control (GNC) system. Performance prediction, commonly used in grain design, calculates the performance curve according to given grain configurations. However, this design method often falls into a trial-and-error approach and fails to obtain the best solution matching the given performance curve. In contrast, the reverse design, a concept rarely mentioned by previous scholars, is an inversed problem of the performance prediction. It seeks the optimal grain shape according to the given performance curve [1]. The relation between the performance prediction and grain reverse design is shown in Figure 1.

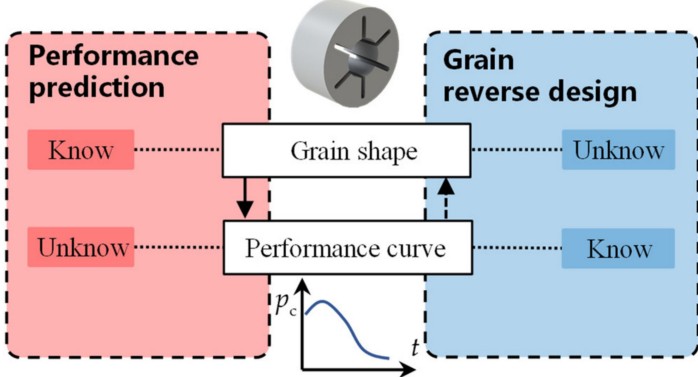

**Figure 1.** Relation between the performance prediction and grain reverse design.

The reverse design is an optimization problem of grain configuration essentially. It is always a nonconvex nonlinear problem, and the objective function does not appear to have a clear analytical formula. Therefore, it is hard to find the optimal global solution via traditional line search methods or trust-region methods. Sforzini [2,3] is the first to study reverse design. The developed solid rocket motor design and optimization (SRMDO) code uses the pattern search method to optimize the 14 parameters of a 3D grain. The objective function is to minimize the square sum of the gap between the designed thrust curve and the given one. Since the 21st century, with the emergence and broad application of heuristic optimization algorithms such as the particle swarm optimization (PSO) and the genetic algorithm (GA), it has become possible to solve the reverse design practically. Albarado [4] uses a hybrid optimizer combining pattern search and particle swarm to minimize the square sum of the gap between the designed pressure curve and the given one. Yücel [5] uses the GA to optimize a finocyl grain to ensure that the error of the thrust curve is minimum. Recently, the reverse design effort is also denoted as the performance matching by Wu [6] or predefined performance criteria by Hashish [7]. So far, the PSO and GA has become a common method for grain optimization design [8–10].

Although researchers have developed all kinds of procedures to achieve reverse design, these procedures all share the same pattern. Firstly, the geometry of the grain is strictly determined by several parameters. Then, the selection of the grain shape, such as the star, wagon wheel, and finocyl shape, is supposed to be consistent with historical experience. After that, the global optimization algorithms are conducted to optimize the geometric parameters so that the designed performance curve is as coherent as possible with the goal. If it fails to find the optimal solution, reselection of the grain shape and redesign are necessary. Consequently, these procedures are semi-empirical and parameter-driven.

The traditional semi-empirical parameter-driven optimization methods have significant shortcomings. Firstly, the design results are limited by the existing grain shapes and historical experience. Although there have been attempts to use expert systems to select grain configurations [11], it is still difficult to design new grain shapes with higher loading fraction and higher performance. Secondly, to prevent interference between the geometric features of the grain, there are numerous nonlinear constraints between the geometric parameters. It is challenging to fully understand these constraints and add them to the optimization algorithm. In addition, the number of optimized variables is equal to the number of geometric parameters, so the degree of freedom of the design is relatively low, making it frequently fail to meet the given performance curve. Finally, the geometric parameters corresponding to different grain shapes are different, so it is not easy to develop a unified optimization framework for multiple grain shapes.

To overcome the shortcomings of the traditional methods of the reverse design, it is necessary to introduce the shape optimization technique. Shape optimization, also considered as a special topology optimization, has been widely used in the mechanical structure design. It is dedicated to finding the optimal distribution of materials instead of

optimal geometric parameters [12–14]. The materials distribution, that is, the propellant distribution, can be defined by the phase field. Conventionally, the phase field function can be expressed by basis functions, such as finite element basis functions, radial basis functions, and spectral parameterization [13]. With the help of machine learning, a simple feedforward neural network (FNN) is also capable of expressing the phase field. Once FNN parameterizes the phase field, one can easily define and manipulate an irregular grain.

Unlike mechanical structure optimization, the challenging part of grain shape optimization lies in the burn-back analysis of the irregular grain shape defined by the phase field. As the proceeding of combustion, the burning surface of the grain changes, which is known as surface regression. The motor's internal ballistic characteristics can only be obtained by accurately calculating the burning surface area. There are two types of burn-back analysis methods: the non-meshing and meshing methods. Non-meshing methods include the analytical method [15,16], general coordinate method [17,18], and CAD modeling method [19,20]. Meshing methods include the minimum distance method [21,22], moving grid method [23,24], level-set method [9,25,26], and fast marching method [27]. On the one hand, the non-meshing methods are accurate, computationally efficient, and reliable. However, they can hardly adapt to the grains with irregular geometry or complex distribution of burning rate. Thus, the non-meshing methods are unfit to apply in shape optimization tasks. On the other hand, the meshing methods possess great generality, but they need a lot of pre-processing and post-processing work. Therefore, they are not yet widely used in engineering. To achieve shape optimization, designers urgently need a new meshing method that is easy to implement on any irregular grain shape—this will be one of the focuses in this work.

This work proposes the shape optimization method based on the evolutionary neural network to achieve the reverse design of two-dimensional (2D) grain. Given the pressure-time curve, the GA is used to optimize the hyperparameters of the neural network which can fully define the grain shape to minimize the gap between the designed results and the goal. As the method uses the concept of shape optimization, it does not need to pre-select any grain shape, and the designers can be free from defining different kinds of geometric parameters for specific grain shapes. Furthermore, the degree of freedom in the evolutionary neural network is controllable, so it will obtain much more optimal and suboptimal solutions than the traditional methods. The method has the potential to apply in many new application scenarios, such as the conceptual design of creative grain shapes, the reconstruction of an actual grain, and the shape design of the additive manufactured grain.

The structure of this work is as follows. The second chapter begins with the steps of grain reverse design. It will continue with a modified PEF method to realize the gas-solid unified burn-back analysis based on the phase field. Then, it will discuss the definition of the phase field by introducing the neural network. Finally, the chapter ends with the genetic algorithm framework of reverse design. The third chapter will provide some test results of the method in detail, and the fourth chapter will give the conclusions.

## 2. Methods

### 2.1. Steps of the Grain Reverse Design

The performance prediction of solid rocket motor can be divided into two steps in practice, as shown on the left side of Figure 2. The first step is the burn-back analysis and the second step is the internal ballistic calculation. Similarly, the grain reverse design, essentially an inversed problem of the performance prediction, can also be divided into two steps, as shown on the right side of Figure 2. The first step, denoted as **reverse internal ballistic calculation**, converts the performance curve ($p_c \sim t$ curve) into variation of burning surface area with burned web thickness ($A_b \sim w$ curve), and the given variables are listed in Table 1. The second step, denoted as **grain reconstruction**, is an optimization process that finds the grain shape matches the $A_b \sim w$ curve most. This division strategy decomposes the reverse design, which can effectively reduce the complexity.

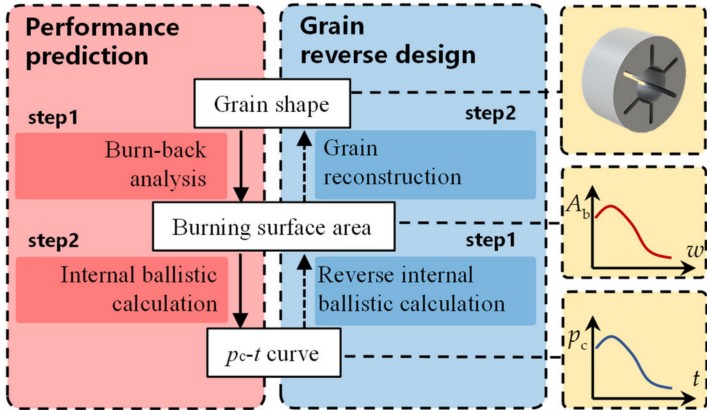

**Figure 2.** Steps of performance prediction and grain reverse design.

**Table 1.** Given variables for reverse internal ballistic calculation.

| Symbol | Notation |
| --- | --- |
| $N$ | Number of geometry units (half an angle) |
| $R$ | Outer radius |
| $\rho_p$ | Propellant density |
| $c^*$ | Propellant characteristic velocity |
| $a$ | Burning rate coefficient |
| $n$ | Pressure index |
| $p_c \sim t$ curve | Pressure-time curve |

2.1.1. Step1: Reverse Internal Ballistic Calculation

To solve the reverse internal ballistic calculation, the following three assumptions are proposed:

**Assumption 1:** The nozzle throat to grain port area ratio of the motor is small enough so that the gas flow speed is low, and there is no erosive combustion phenomenon.

**Assumption 2:** The combustion chamber pressure does not change drastically during the whole working process, which makes the equilibrium pressure formula reasonable.

**Assumption 3:** The characteristic velocity and density of the propellant do not vary with the combustion chamber pressure.

For brevity, the concept of similar grains is proposed. As shown in Figure 3, two 2D grains are admitted being similar if all their sides are proportional. The dimensionless burn perimeter $\hat{l}_b$ is defined as Equation (1):

$$\hat{l}_b = \frac{l_b}{R} \tag{1}$$

where $R$ is the outer radius of the grain. For 2D grains, the burn perimeter $l_b$ is defined as Equation (2):

$$l_b = \frac{A_b}{NL} \tag{2}$$

where $A_b$ is the burning surface area of the grain, and $L$ is the length of the grain. It is worth noting that the grain shape can often be divided into the several minor geometry units that can repeat themselves by rotational and mirror transformations. Therefore, concentrating on one single unit provides much more efficiency than the whole grain. The number of units (half an angle) is denoted as $N$. For example, in Figure 3, one geometric unit has half a slot, yielding $N = 12$.

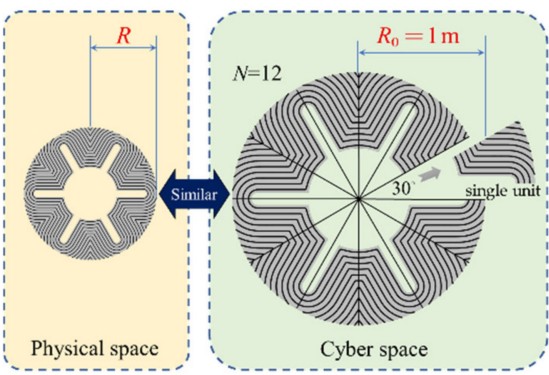

**Figure 3.** Illustration of the similar grains.

The dimensionless burned web thickness $\hat{w}$ is also defined as Equation (3):

$$\hat{w} = \frac{w}{R} \tag{3}$$

For 2D grains, the reverse internal ballistic calculation needs to convert the known $p_\text{c} \sim t$ curve into the unknown $\hat{l}_\text{b} \sim \hat{w}$ curve. As shown in Figure 3, although the grain in the physical space may be much smaller (or greater) than the one in the cyber space, they are still similar grains and sharing the same $\hat{l}_\text{b} \sim \hat{w}$ curve.

The advantage of nondimensionalization is that it can significantly simplify the second step of reverse design, denoted as grain reconstruction. Any grain in physical space can be transformed into a standard grain (outer radius $R_0$ is 1 m) in the cyber space through scaling transformation, so the burn-back analysis can be performed within a unified standard framework. The designers only need to focus on optimizing the internal profile without considering a mutable outer radius. As soon as the grain in the cyber space meets the requirements, it can be directly returned to the physical space through simple scaling transformation.

The calculation formula of the burn rate $r_\text{b}$ can usually be expressed as Equation (4) [28]:

$$r_\text{b} = a p_\text{c}^n \tag{4}$$

where $a$ and $n$ are the burn rate coefficient and pressure index, respectively. Derived from the steady-state continuity equation, the equilibrium pressure of the combustion chamber is Equation (5):

$$p_\text{c}(t) = \left( \rho_\text{p} c^* a \frac{N L l_\text{b}}{A_\text{t}} \right)^{\frac{1}{1-n}} \tag{5}$$

where $\rho_\text{p}$, $c^*$ and $A_\text{t}$ are the propellant density, characteristic velocity, and nozzle throat area, respectively. According to Equations (4) and (5), the expression for $\hat{l}_\text{b}$ and $\hat{w}$ over time is as Equation (6):

$$\begin{cases} \hat{l}_{\text{b,T}}(t) C_\text{T} = \frac{p_\text{c}^{1-n}(t)}{N \rho_\text{p} c^* a} \\ \hat{w}_\text{T}(t) = \frac{\int_0^t a p_\text{c}^n(\tau) \, d\tau}{R} \end{cases}, \ (0 \le t \le t_\text{a}) \tag{6}$$

where $t_\text{a}$ is the action time, and the subscript T in the formula represents the design target, which is the requirement of designers. The $C_\text{T}$ is a dimensionless variable expressed as Equation (7):

$$C_\text{T} = L R / A_\text{t} \tag{7}$$

Since the designers are uncertain about the $L$ and $A_\text{t}$ from the beginning, the $C_\text{T}$ is also a quantity that needs to be designed. $\hat{l}_{\text{b,T}}(t) C_\text{T}$ is the product of $\hat{l}_{\text{b,T}}(t)$ and $C_\text{T}$. It is an independent variable and can also be regarded as a dimensionless burn perimeter. It is important to emphasize that $N$ is unknown in practice, but to reduce the computation cost,

$N$ is considered to be a given integer in this work. Since $N$ is a finite integer, the optimal $N$ can be found by enumeration.

Given the variables in Table 1, the $\left(\hat{l}_{b,T}C_T\right) \sim \hat{w}_T$ curve will be obtained via Equation (6). So far, the reverse internal ballistic calculation is completed.

### 2.1.2. Step2: Grain Reconstruction

Given the $\left(\hat{l}_{b,T}C_T\right) \sim \hat{w}_T$ curve, grain reconstruction refers to finding a 2D grain that satisfies this curve. The following three requirements are proposed:

**Requirement 1:** The burning starts from the internal port surface of the grain. This kind of grain is known as internal burning grain.

**Requirement 2:** The solid region for one geometric unit shall be connected without holes.

**Requirement 3:** The gas-solid interface can be an irregular curve.

Using the subscript D to represent the design result, the grain reconstruction is to optimize the internal profile in the cyber space so that the $\hat{l}_{b,D} \sim \hat{w}_D$ curve obtained by the burn-back analysis matches the $\hat{l}_{b,T} \sim \hat{w}_T$ curve. In practice, a method based on linear regression can evaluate the matching degree quantitatively according to the $\hat{l}_{b,D} \sim \hat{w}_D$ curve and $\left(\hat{l}_{b,T}C_T\right) \sim \hat{w}_T$ curve calculated from Equation (6) (see Section 2.4.2 for details). Figure 4 is an example of matching the performance. The design result in Figure 4a has a higher matching degree than Figure 4b. The average value of the ratio $\hat{l}_{b,T}C_T/\hat{l}_{b,D}$, denoted as $C_D$, can be used as the design result of $LR/A_t$.

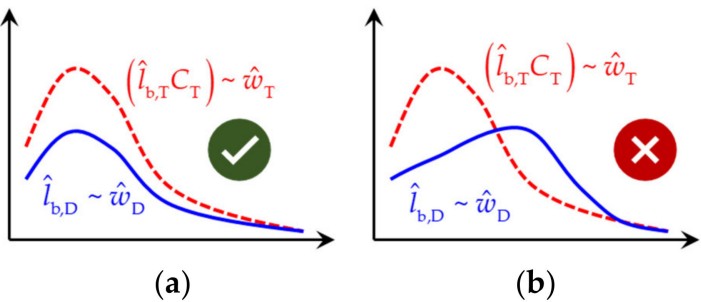

**Figure 4.** Illustration of matching the performance (**a**) acceptable and (**b**) unacceptable.

So far, the procedure of the reverse design is clear. The specific methods will be described further in the next sections.

### 2.2. Burn-Back Analysis of Irregular Grain Shapes

### 2.2.1. PEF Method (Poisson Equation—Eikonal Equation—Finite Element Method)

Since the method for burn-back analysis in this work uses the Poisson equation, the eikonal equation, and the finite element method, it is called the PEF method. Similar to the propagation of light [29], burning surface regression of complex grains can be described by the eikonal equation. In the grain with uniform burning rate, it can be written as Equation (8):

$$\nabla W \cdot \nabla W = 1 \tag{8}$$

where $W$, in meters, is the field of burned web thickness $w$, which gives the thickness of the burned web corresponding to any spatial coordinate. Any isosurface of the $W$ field represents the burning surface. The meaning of Equation (8) is that the gradient of the $W$ field is 1 everywhere, which is consistent with the assumption of parallel-layer combustion widely adopted in solid rocket motor design.

There are two kinds of boundary conditions, that is, the Dirichlet boundary condition $W = 0$ at the initial burning surface and the zero-flux boundary condition $\partial W/\partial n = 0$ at the flame-retardant surface or symmetry surface. However, for the grain with the non-convex gas cavity, as the burning surface moves, the $W$ field is no longer derivable

at the intersection of burning surfaces (see Figure 5). One possible way to ensure that the burning surface keeps smooth numerically is to introduce a burning rate that varies with the curvature. The essence is to add a diffusion term based on Equation (8). Then, Equation (9) is obtained [30]:

$$\alpha \nabla^2 W = \nabla W \cdot \nabla W - 1 \tag{9}$$

where $\alpha$ is the diffusion term coefficient, a positive number in meters.

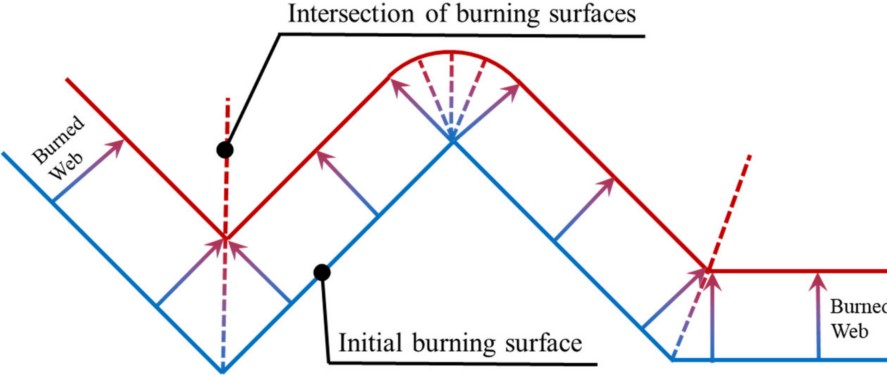

**Figure 5.** Illustration of burning surface regression.

If the diffusion term coefficient is too large, the calculation accuracy will decrease. On the other hand, if it is too small, the convergence of the finite element method will deteriorate. The recommended value range is:

$$\alpha = (0.1 \sim 0.2)\delta l \tag{10}$$

where $\delta l$ is the local element size. Figure 6 shows the results of burning surface regression for a typical star grain ($R = 0.5$ m, $N = 12$) with different meshes. Figure 7 shows that with the refinement of the element size, compared with the analytic solution, the accuracy of PEF method will continue improving.

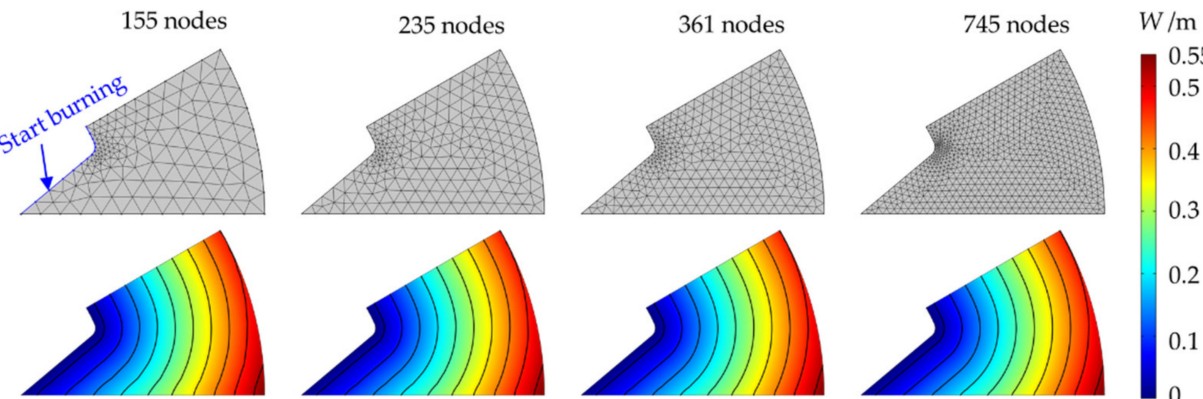

**Figure 6.** Results of burning surface regression for the typical star grain with different meshes.

Compared to the traditional method of burn-back analysis, the advantages of the PEF method are significant. It can perform the burn-back analysis of 3D grain with irregular geometry robustly. In addition, since the PEF method converts the burn-back analysis into an analogous steady heat conduction problem, it can be directly modeled and solved using COMSOL, ANSYS, ABAQUS, and other commercial finite element software, which are relatively easy to implement.

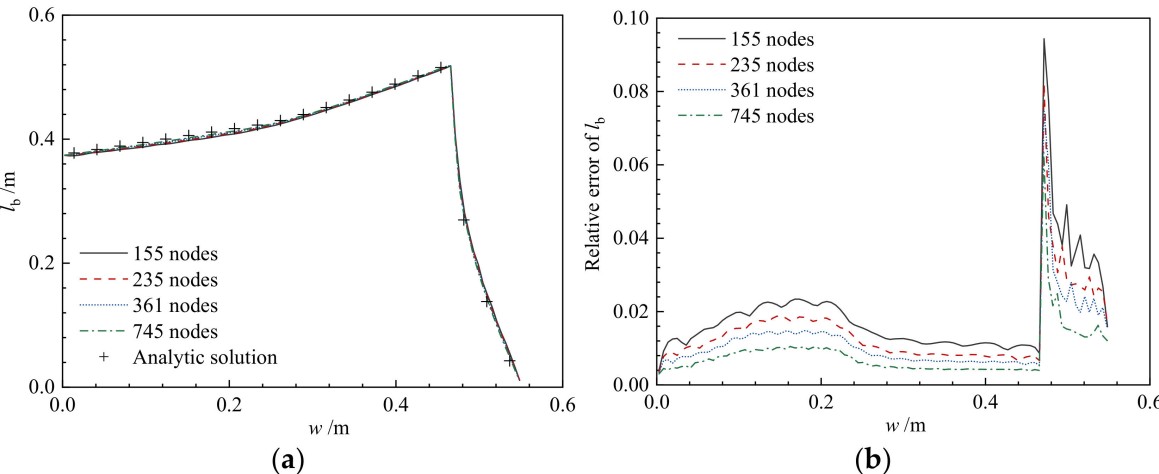

**Figure 7.** Results of burn perimeter for the typical star grain with different meshes: (**a**) Relation between the burn perimeter and the web thickness and (**b**) Relation between the relative error of burn perimeter and the web thickness.

2.2.2. Modified PEF Method ($\varphi$—PEF)

It is worth noting that the solution domain of the PEF method is only the solid-phase region, excluding the gas-phase region. Therefore, the gas-solid unified burn-back analysis cannot be accomplished. Since the exact shape of the solid-phase region is unsure in the current reverse design, the PEF method must be modified.

The gas-solid unified burn-back analysis can be established by introducing the virtual burning rate and the source term factor. Rewrite the governing equation as Equation (11):

$$\alpha r \nabla^2 W = \left( r^2 \nabla W \cdot \nabla W - 1 \right) s \tag{11}$$

where $r$ is the dimensionless virtual burning rate, and $s$ is the dimensionless source term factor.

Inspired by the level-set method [31] and phase field method [32] of structural topology optimization, the concept of phase field is then introduced to clearly describe the gas-phase region, solid-phase region, and their interfaces. A dimensionless scalar $\varphi(x,y)$ is defined as Equation (12):

$$\text{Phase} = \begin{cases} \text{Solid,} & \varphi(x,y) > 0 \\ \text{Interface,} & \varphi(x,y) = 0 \\ \text{Gas,} & \varphi(x,y) < 0 \end{cases}, \; (-1 \le \varphi(x,y) \le 1) \tag{12}$$

The distribution of $\varphi(x,y)$ in the whole domain constitutes the phase field. As long as this distribution is obtained, it is equivalent to obtain a 2D grain shape. The initial burning surface is given by the zero level $\varphi(x,y) = 0$.

Based on the phase field, the virtual burning rate field $r$ can be constructed, subsequently, which is defined as Equation (13):

$$r = \begin{cases} 1, & \varphi > 0, \\ 1 + K\varphi^2, & \varphi \le 0 \end{cases} \tag{13}$$

where $K$ is a constant much larger than 1. The physical meaning of the virtual burning rate $r$ is that the burning rate in the gas region is so large that the web thickness field $W$ is approximately equal at any position.

Unfortunately, when the virtual burning rate $r$ presents in the gas-phase region, the source term near the gas-solid interface has a great jump. The iteration of finite element

method (FEM) is difficult to converge due to its nonlinearity. Therefore, the source term factor *s* is then introduced, which is defined as Equation (14):

$$s = \begin{cases} 1, & \varphi > 0 \\ 0, & \varphi \leq 0 \end{cases} \tag{14}$$

The physical meaning of the source term factor is as follows. In the solid-phase region, the original governing equation is still maintained. While in the gas-phase region, the source term of the governing equation is set to 0, and the governing equation degenerates into a steady-state heat conduction equation without any internal heat source, also known as the Laplace equation. Solving the Laplace equation in the gas-phase region will significantly improve the convergence of the FEM.

Since the method uses phase field, it is called the $\varphi$—PEF method in this work. It has many advantages. Firstly, the burn-back analysis can be completed by using a fixed mesh without any extra work to track the moving interface. It avoids remeshing after changing the grain shape, which is quite useful in the shape optimization. Secondly, since the mesh is fixed, setting up and defining Dirichlet boundary conditions are simplified. Finally, inheriting the advantages of the PEF method, it can be directly modeled and solved in the commercial finite element software, which is relatively easy to implement.

To compare the difference between the PEF method and $\varphi$—PEF method, the calculation results for the same grain are illustrated in Figure 8. The solid-region of the grain is formed by the intersection of a square with a side length of 2 m and a circle with a radius of $\sqrt{5/2}$ m. The workflow of the PEF method is as follows. Firstly, the geometry is drawn using CAD software. The right-hand-side boundary condition is set to the flame-retardant layer and the rest of the boundary are set to the Dirichlet boundary condition. Then with the help of unstructured mesh, Equation (9) is discretized into an algebraic equation to solve the *W* field, while, the $\varphi$—PEF method uses the phase field to describe the shape of the solid region of the grain in a fixed square region, and the phase field for this problem has an analytical expression as Equation (15):

$$\varphi(x,y) = 1 - \frac{2}{5}\left((x-1)^2 + y^2\right) \tag{15}$$

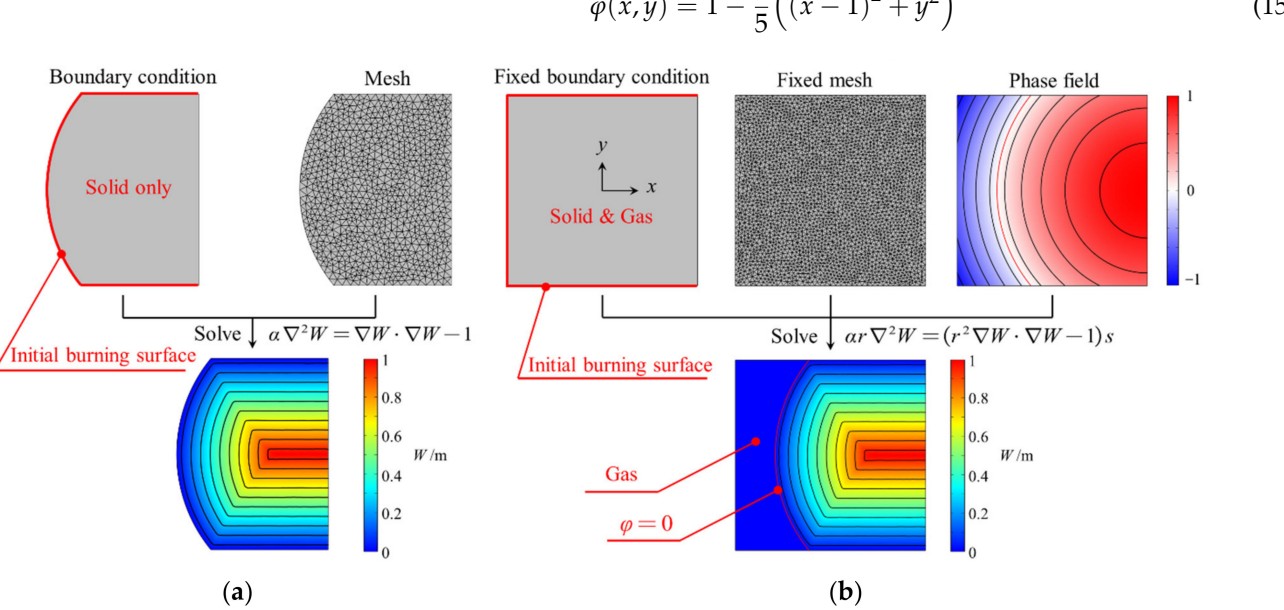

**Figure 8.** Illustration of burn-back analysis methods: (**a**) PEF method and (**b**) $\varphi$—PEF method.

The *W* field is obtained by solving the algebraic equation derived from Equation (11). It can be seen from Figure 8 that the results obtained by both methods are identical.

Once the *W* field is obtained, the burning surfaces can be obtained by extracting the isosurfaces of the *W* field, and the burn perimeter $l_b$ can be calculated from these burning

surfaces. Since the $\varphi$—PEF method is much more flexible, we chose it to achieve the burn-back analysis of the irregular grain shape.

### 2.3. Defining the Phase Field Using Neural Network

2.3.1. The Direct Method to Solve the Optimization Problem

In Section 2.2, the grain design problem is transformed into the design work of a continuous two-dimensional function $\varphi(x,y)$, and this function will become the optimization variable. The corresponding optimal solution can be obtained by using indirect methods (IM) such as optimal control theory [33]. Although the IM can effectively guarantee optimality and high accuracy, there are still great difficulties in deriving the first-order optimal conditions and solving two-point boundary value problems [34]. Therefore, this work will focus on the more practical method denoted as the direct method (DM) to solve the optimization problem.

The DM, approximating the function to be solved as a nonlinear combination of finite parameters, transforms the design problem of the function into a multi-parameter optimization problem [35]. Finally, the infinite-dimensional optimization problem is transformed into a finite-dimensional one, as shown in the Equation (16):

$$\varphi(x,y) \rightarrow \hat{\varphi}(\boldsymbol{p},x,y) \tag{16}$$

where $\boldsymbol{p}$ is the independent vector of the optimization problem, $\hat{\varphi}$ is a predetermined function and is often selected as a linear combination of basis functions. However, its accuracy is generally insufficient especially for nonlinear problems, which cannot meet the demands for an engineering application, so the neural network is proposed next.

2.3.2. Design of Feedforward Neural Network

Artificial neural network (ANN) refers to a series of mathematical models inspired by biology and neuroscience. These models can simulate the biological neural network by mathematically abstracting the biological nervous system. After decades of development, ANN, the backbone of artificial intelligence technology, has been widely used in mathematical and engineering fields such as fitting, classification, and clustering.

Feedforward neural network (FNN) is the first ANN invented. In each layer, neuron units can receive signals from units in the previous layer and generate signals to transmit to the next layer. According to the Universal Approximation Theorem, for an FNN with a linear output layer and at least one hidden layer using a nonlinear activation function, as long as the number of neurons in the hidden layer is large enough, the FNN can approximate any bounded closed set function defined in the real number space with any accuracy [36,37].

After considering the computational cost and fitting accuracy, we designed an FNN of one single hidden layer to construct the function $\hat{\varphi}(\boldsymbol{p},x,y)$. The input layer is denoted as layer 0, the hidden layer is denoted as layer 1, and the output layer is denoted as layer 2. Table 2 gives the relevant notation of the FNN used in this work.

**Table 2.** Notation of the symbols in the FNN.

| Symbol | Notation |
|---|---|
| $M_l$ | Number of units in layer $l$ |
| $f_l(\cdot)$ | Activation function of the units in layer $l$ |
| $\boldsymbol{W}^{(l)} \in \mathbb{R}^{M_l \times M_{l-1}}$ | Weight matrix from layer $l$-1 to layer $l$ |
| $\boldsymbol{b}^{(l)} \in \mathbb{R}^{M_l}$ | Bias vector from layer $l$-1 to layer $l$ |
| $\boldsymbol{z}^{(l)} \in \mathbb{R}^{M_l}$ | Net input (net activation) of units in layer $l$ |
| $\boldsymbol{a}^{(l)} \in \mathbb{R}^{M_l}$ | Output (activation) of units in layer $l$ |

If $\boldsymbol{a}^{(0)} = (x, y)^{\mathrm{T}}$, the FNN can propagate information by iterating Equation (17):

$$\begin{aligned} \boldsymbol{z}^{(l)} &= \boldsymbol{W}^{(l)}\boldsymbol{a}^{(l-1)} + \boldsymbol{b}^{(l)}, \\ \boldsymbol{a}^{(l)} &= f_l(\boldsymbol{z}^{(l)}) \end{aligned} \qquad (l = 1, 2) \qquad (17)$$

First, the net activation of layer $l$ is calculated according to the activation of layer $l$-1 and the weight matrix and bias vector of layer $l$. Then, the activation of layer $l$ is calculated according to the nonlinear activation function. Equation (17) can be written in a compact form:

$$\boldsymbol{a}^{(l)} = f_l(\boldsymbol{W}^{(l)}\boldsymbol{a}^{(l-1)} + \boldsymbol{b}^{(l)}) \qquad (18)$$

According to several tests, setting $M_1 = 20$ can provide enough design freedom without significantly increasing the computational burden of the optimization. The parameters of the FNN structure used in this study are shown in Table 3. So far, the specific structure of the FNN has been determined, as shown in Figure 9.

**Table 3.** Parameters of the FNN structure.

| Items | Value |
|:---:|:---:|
| $M_0$ | 2 |
| $M_1$ | 20 |
| $M_2$ | 1 |
| $f_1(z_i^{(1)})$ | $\dfrac{2}{1+e^{-2z_i^{(1)}}} - 1$ |
| $f_2(z_i^{(2)})$ | $\dfrac{2}{1+e^{-2z_i^{(2)}}} - 1$ |

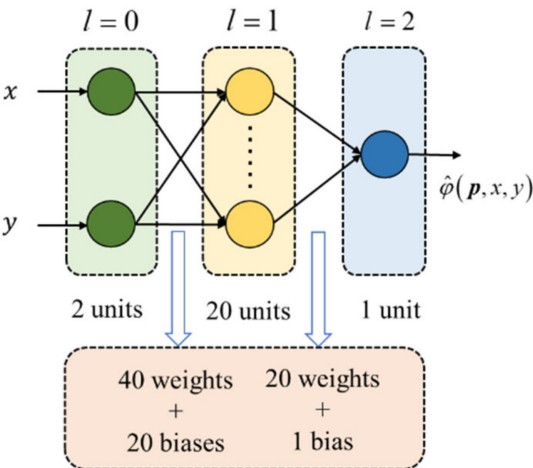

**Figure 9.** Structure diagram of the FNN.

The total number of tunable hyperparameters of the neural network can be found in Equation (19):

$$N_{\mathrm{hp}} = (M_0 + 1)M_1 + (M_1 + 1)M_2 \qquad (19)$$

Substituting the data in Table 3 yields $N_{\mathrm{hp}} = 81$. After comparing with Equation (16), it can be seen that the vector $\boldsymbol{p}$ is an 81-dimensional vector, and its elements are composed of all the elements in the matrices $\boldsymbol{W}^{(1)}$, $\boldsymbol{W}^{(2)}$ and the vectors $\boldsymbol{b}^{(1)}$, $\boldsymbol{b}^{(2)}$.

### 2.3.3. Pre-Training Based on Supervised Learning

To obtain faster optimization speed, an appropriate initial guess of vector $p_0$ needs to be selected according to specific prior knowledge to transform the problem into optimizing correction $\Delta p$, as shown in Equation (20):

$$\hat{\varphi}(p, x, y) \rightarrow \widetilde{\varphi}(p_0, \Delta p, x, y), \ p = p_0 + \Delta p \qquad (20)$$

where the range of $\Delta p$ is chosen to be $[-1, 1]$ to avoid holes in the grain. This technique is denoted as pre-training in this work. In the rest of this section, $p_0$ will be obtained via supervised learning from a typical tube grain shape.

The phase field of a typical tube grain shape can be written in analytical form. Then, the position coordinates and phase field of the grain are taken as training samples, which can be expressed as Equation (21):

$$\text{sample set} = \left\{ \left[ (x^{(n)}, y^{(n)}), \varphi_0^{(n)} \right] \right\}_{n=1}^{M} \qquad (21)$$

where $M$ is the total number of the samples and $\varphi_0^{(n)}$ is the phase field of the typical tube grain shape.

The loss function in the form of mean square error (MSE) is defined as follows:

$$\sigma_{\text{MSE}} = \frac{\sum\limits_{n=1}^{M} \left[ \varphi_0^{(n)} - \hat{\varphi}\left( p, x^{(n)}, y^{(n)} \right) \right]^2}{M} \qquad (22)$$

where $\hat{\varphi}\left( p, x^{(n)}, y^{(n)} \right)$ represents the estimated value of the phase field.

This work obtains the corresponding parameter vectors using the Bayesian regularization training method [38]. This method effectively avoids overfitting and has an excellent fitting effect for complex problems. The variation curve of the loss function with the training epoch is shown in Figure 10. The output error of the neural network is shown in Figure 11. The mapped phase field of the tube grain is shown in Figure 12.

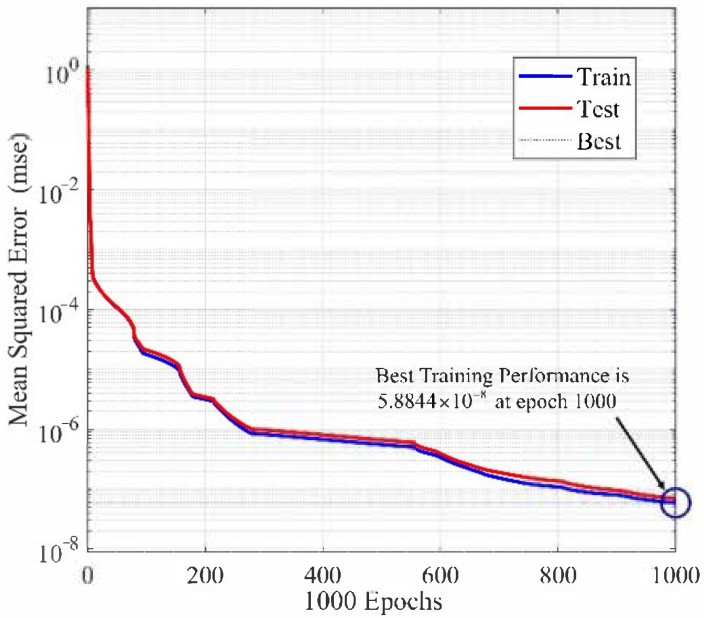

**Figure 10.** Relation between the MSE and the training epoch.

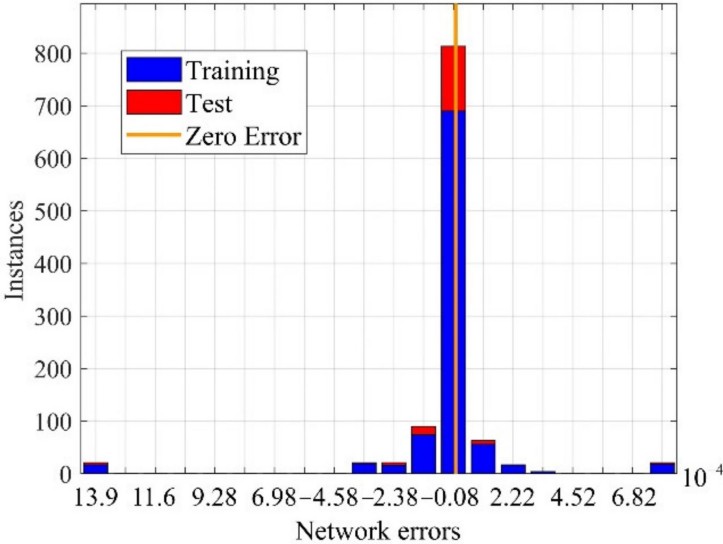

**Figure 11.** Error distribution of the neural network output.

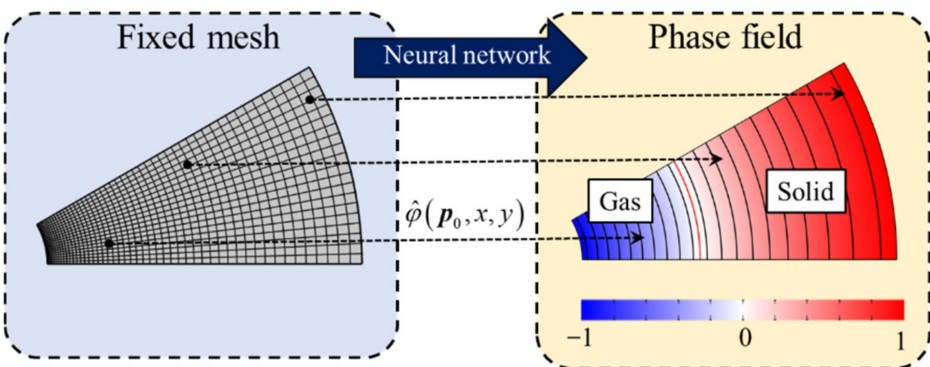

**Figure 12.** Mapped phase field of the tube grain using the pre-training neural network.

As shown in Figure 10, the MSE decreases monotonically with the training epoch. When the training epoch reached 1000, the MSE reached the minimum value $5.8844 \times 10^{-8}$. In addition, the variation of MSE between the training set and the test set is relatively small, indicating that no overfitting has occurred in the training process. It can be seen from Figure 11 that the output error of the network is at a small level. In summary, the pre-training can obtain relatively accurate initial parameter values $\boldsymbol{p}_0$.

### 2.4. Optimizing the Neural Network Using the Genetic Algorithm

#### 2.4.1. Genetic Algorithm Framework

The genetic algorithm (GA) uses neither sensitivity derivatives nor reasonable starting points. As a direct search-based method, it can solve the global optimization problems with a non-analytical objective function, which is suitable for optimizing hyperparameters of neural networks. Therefore, in this work, GA is used to optimize the neural network to achieve a phase field matching the performance best. The neural network combined with GA is also known as the evolutionary neural network.

Figure 13 illustrates the framework of the GA, where the optimized variable is the $\Delta \boldsymbol{p}$ in Equation (20), containing a total number of 81 hyperparameters. The next section will focus on the computation of the objective function of GA.

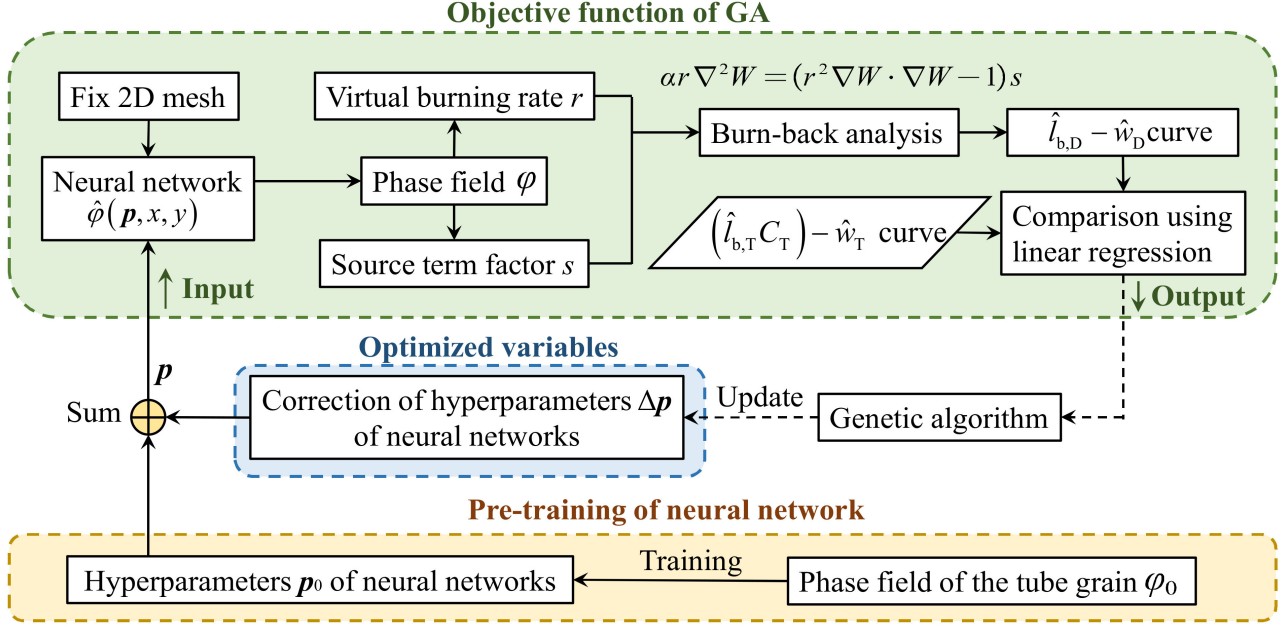

**Figure 13.** Genetic algorithm framework.

2.4.2. Objective Function

Determining a suitable objective function is one of the most challenging tasks in the reverse design. Using different objective functions will lead to different results. The objective function shall compare the similarity between $\hat{l}_{b,D} \sim \hat{w}_D$ curve and $\left(\hat{l}_{b,T}C_T\right) \sim \hat{w}_T$ curve, and output the performance-matching degree (see Figure 4).

To evaluate the similarity between curves, a method based on linear interpolation combined with linear regression is proposed. For the convenience of writing, the function obtained by linear interpolation of $\hat{l}_{b,D} \sim \hat{w}_D$ is denoted as $f_1$, and the function obtained by linear interpolation of $\left(\hat{l}_{b,T}C_T\right) \sim \hat{w}_T$ is denoted as $f_2$. Take $n$ nodes at equal intervals within the value range of the independent variable to form a vector $\boldsymbol{x}$:

$$\boldsymbol{x} = [x_1, x_2, \cdots, x_n]^{\mathrm{T}} \tag{23}$$

The vectors $\boldsymbol{y}_1$ and $\boldsymbol{y}_2$ is calculated using $f_1$ and $f_2$, and the formula is as follows:

$$\begin{aligned} \boldsymbol{y}_1 &= [f_1(x_1), f_1(x_2), \cdots, f_1(x_n)]^{\mathrm{T}}, \\ \boldsymbol{y}_2 &= [f_2(x_1), f_2(x_2), \cdots, f_2(x_n)]^{\mathrm{T}} \end{aligned} \tag{24}$$

If $f_1$ and $f_2$ are approximately proportional, then there is:

$$\boldsymbol{y}_1 C_D = \boldsymbol{y}_2 \tag{25}$$

Equation (25) contains one unknown variable $C_D$ and $n$ equations. Since the number of equations is much larger than the number of unknowns, it is overdetermined and generally has no solution. However, the linear regression method can be used to find the solution in the presence of the smallest residual error, that is, the least squared solution. The unknown variable $C_D$ is approached using Equation (26):

$$C_D = \frac{\boldsymbol{y}_1^{\mathrm{T}} \boldsymbol{y}_2}{\boldsymbol{y}_1^{\mathrm{T}} \boldsymbol{y}_1} \tag{26}$$

According to the discussion in Section 2.1.2, $C_D$ can be used as the design result of $LR/A_t$. In order to quantitatively evaluate the similarity between $f_1$ and $f_2$, the coefficient of determination ($COD$) can be calculated as Equation (27):

$$COD = 1 - \frac{(y_2 - C_D y_1)^T (y_2 - C_D y_1)}{(y_2 - \bar{y}_2)^T (y_2 - \bar{y}_2)} \tag{27}$$

where $\bar{y}_2$ is an n-dimensional vector consisting of the mean value of $y_2$. The $COD$ is a number never greater than 1. The closer it is to 1, the more similar are the $f_1$ and $f_2$. When $COD$ is equal to 1, Equation (25) will be strictly satisfied. Since the GA algorithm always seeks the smallest objective function, we have Equation (28):

$$FV = 1 - COD \tag{28}$$

where $FV$ is the abbreviation of Fitness Value. In summary, the goal of the GA algorithm is to minimize Equation (28).

2.4.3. Consideration of Constraints

Constraints can be added to GA in the form of penalty functions. The specific implementation is to add a very large number to the objective function if the current solution violates the constraints, shown as Equation (29):

$$FV = 1 - COD + P \tag{29}$$

where $P$ is the penalty functions. There are two types of constraints that need to be emphasized.

1. Rationality of the grain structure.

As shown in Figure 14, if the grain is completely separated from the chamber wall or there are holes in the grain, it is unacceptable. In practice, holes can be basically eliminated by choosing the range of $\Delta p$ to be $[-1, 1]$. If for every $x \in$ wall there is $\varphi < 0$, then it is known that the grain is completely separated from the chamber wall, and the objective function should be penalized for these cases.

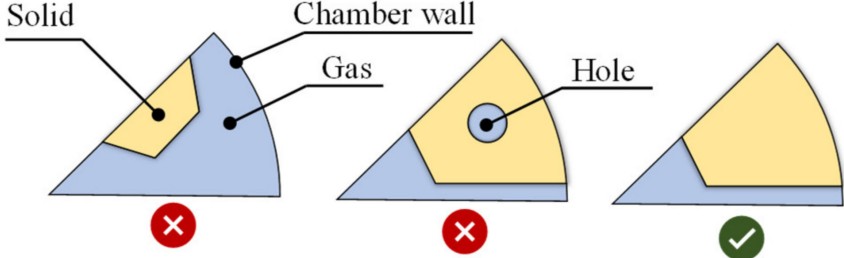

**Figure 14.** Possible grain structure generated by the neural network.

2. Loading fraction constraints.

$C_D$ in Equation (26) is the design result of $LR/A_t$. It can qualitatively describe the loading fraction. The loading fraction decreases with the increase of $C_D$. In order to obtain a more reasonable loading fraction in engineering, $C_D$ range should be selected according to the actual needs. The objective function should be penalized for cases that violate the range.

So far, the method of grain reconstruction is established. The reverse design can be achieved via two separate steps, including reverse internal ballistic calculation and grain reconstruction.

## 3. Results and Discussion

This chapter will give three examples of shape optimization. Sections 3.1 and 3.2 are the reconstruction of given grains, excluding the reverse internal ballistic calculation. Starting from the $\left(\hat{l}_{b,T}C_T\right) \sim \hat{w}_T$ curves of given grains, they are dedicated to finding results most matching the given grains. Based on Sections 3.1 and 3.2, a complete reverse design, including two steps denoted as the reverse internal ballistic calculation and the grain reconstruction, is conducted in Section 3.3. The purpose is to find the grains most matching the given $p_c \sim t$ curve without any additional prior knowledge.

### 3.1. Reconstruction of a Star Grain

#### 3.1.1. Design Target

Star shape is one of the most fundamental grain shapes. In this section, starting from the $\left(\hat{l}_{b,T}C_T\right) \sim \hat{w}_T$ curve of a provided star grain, the reconstruction of the grain is conducted with the evolutionary neural network. The geometry of the provided star grain is shown in Figure 15. The number of the geometry units is 12, and the throat area is $5 \times 10^{-2}$ m.

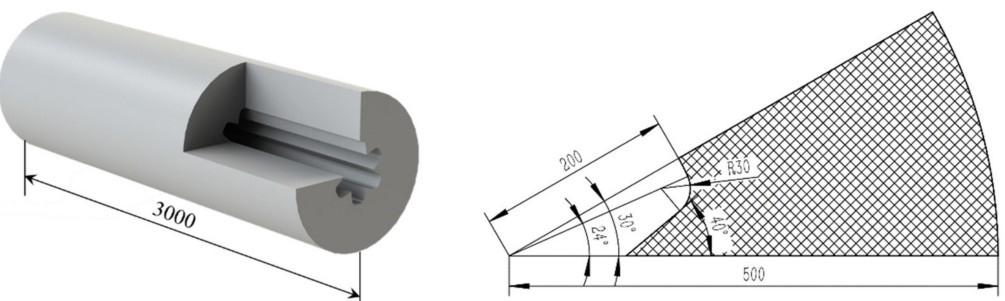

**Figure 15.** Geometry of the given star grain.

The PEF method is used to obtain the $\hat{l}_{b,T} \sim \hat{w}_T$ curve. Considering Equation (7), there is Equation (30):

$$C_T = \frac{LR}{A_t} = 30 \tag{30}$$

Then the $\left(\hat{l}_{b,T}C_T\right) \sim \hat{w}_T$ curve is calculated and plotted in Figure 16.

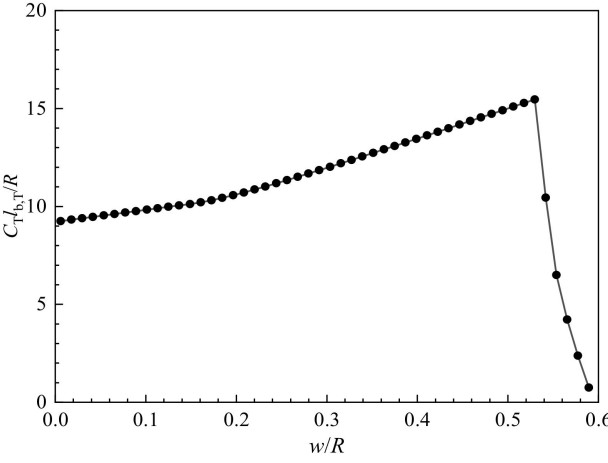

**Figure 16.** Relation between the dimensionless burn perimeter and the dimensionless web thickness of the given star grain.

### 3.1.2. Design Result

Firstly, the computational domain is restricted to a fan shape which is $1/N$ of a whole circle with an inner radius of $0.2R$ and an outer radius of $R$. The initial combustion surface is set to be a circular arc at the inner radius, and the rest of the boundaries are flame retardant. Then, the computational domain is divided into 1000 s-order quadrilateral elements, as shown in Figure 17.

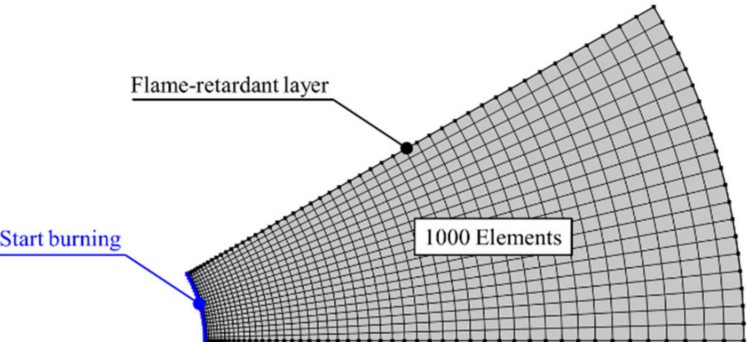

**Figure 17.** Mesh of the computational domain.

Secondly, a phase field is defined using the typical tube grain. According to this phase field, the pre-training of the neural network is completed, as shown in Figure 12.

Thirdly, taking Equation (29) as the objective function, setting the population to 200 and generations to 50, the neural network is evolved via the GA algorithm. If the reduction rate of the objective function is negligible, it can be considered that the optimal shape of the grain is obtained.

The evolutionary process in Figure 18 shows that the shape of the grain changes as the GA algorithm proceeds. The figure presents the best individuals in populations during generations. As the result of pre-training, G0 is a typical tube grain. After one generation of reproduction, G1 tends to be a star grain. However, the shape of G5 appears to be highly irregular. After G10, the irregular shape vanishes, and a continuous and smooth star grain is gradually formed.

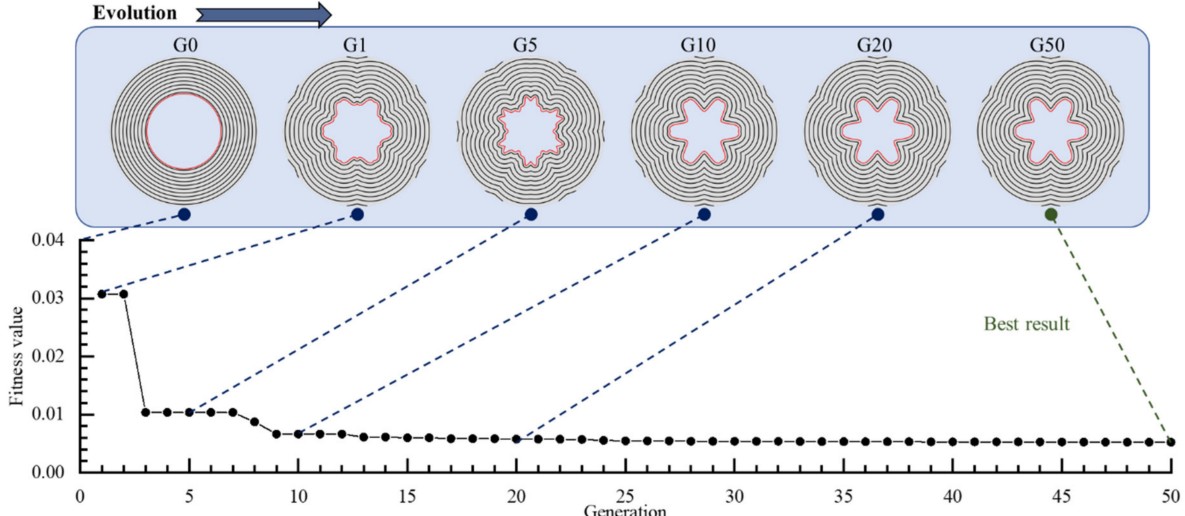

**Figure 18.** Evolution of the designed star grain.

Using G50 as the final design result, linear regression is used to evaluate the gap between the design result and the goal, as shown in Figure 19.

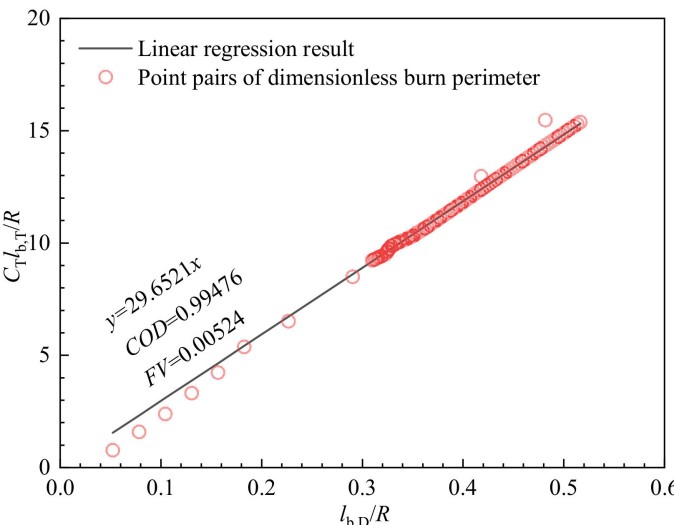

**Figure 19.** Linear regression to evaluate the performance-matching degree of the star grain.

The linear regression is indicated in Equation (31):

$$\hat{l}_{b,T}C_T \approx \hat{l}_{b,D}C_D, \; C_D = 29.6521 \tag{31}$$

Comparing $C_D$ with $C_T$, given in Equation (30), the relative error is calculated in Equation (32):

$$\varepsilon = \frac{C_D - C_T}{C_T} = -1.16\% \tag{32}$$

Since $R = 0.5$ m, the designed grain must satisfy Equation (33):

$$\frac{L_D}{A_{t,D}} = 59.3042 \; \text{m}^{-1} \tag{33}$$

Equation (33) indicates that the length of the grain strictly corresponds to the throat area. Particularly, when $A_{t,D} = 5 \times 10^{-2}$, the designed grain replicates the given one precisely. Furthermore, the dimensionless burn perimeter is plotted in Figure 20 as a function of dimensionless web thickness. Although the tail-off stage has few computational points, the linear interpolation in Equation (24) can ensure the accuracy of the objective function.

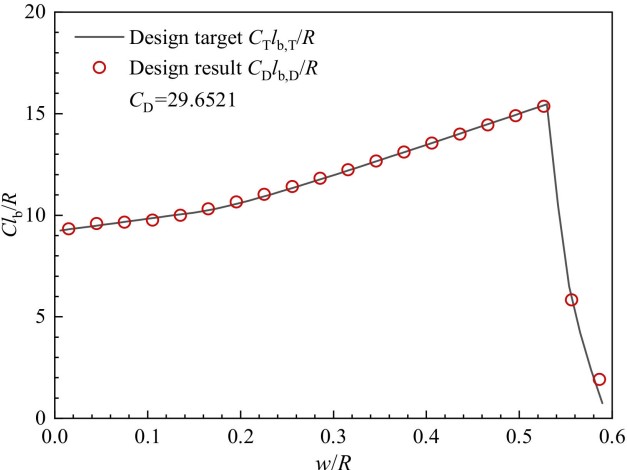

**Figure 20.** Relation between the dimensionless burn perimeter and dimensionless web thickness of the designed grain.

After considering the combustion characteristics of the propellant in Table 4, the combustion chamber pressure is plotted in Figure 21 as a function of time. If the nozzle area ratio and ambient pressure are known, the thrust-time curve can also be obtained.

**Table 4.** Combustion characteristics of the star grain propellant.

| Symbol | Name | Value | Unit |
|--------|------|-------|------|
| $\rho_P$ | Propellant density | 1700 | kg/m$^3$ |
| $c^*$ | Propellant characteristic velocity | 1400 | m/s |
| $a$ | Burning rate coefficient | $2 \times 10^{-4}$ | m/(s-Pa$^{0.3}$) |
| $n$ | Pressure index | 0.3 | — |

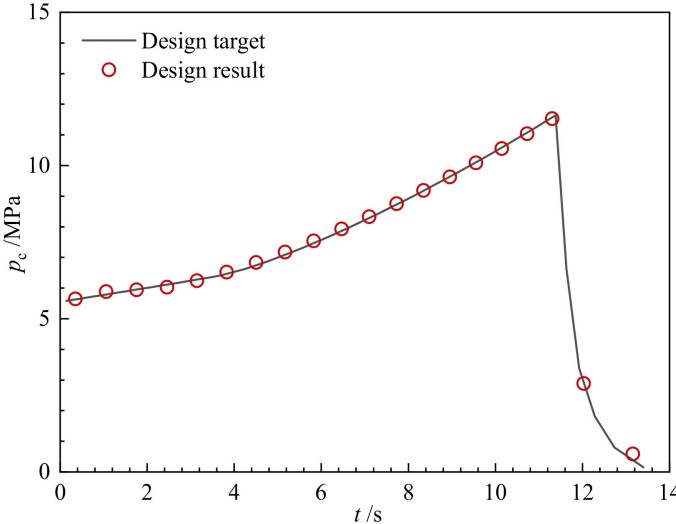

**Figure 21.** Relation between the combustion chamber pressure and the burning time of the designed grain.

It can be seen from the figures that the design result has matched the target curve well. Consequently, the shape optimization method can precisely reconstruct a given star grain.

### 3.2. Reconstruction of a Slotted-Tube Grain

3.2.1. Design Target

Theoretically, the GA algorithm can easily find the global optimal solution of the hyperparameters in the neural network. However, designers often need to consider several constraints such as loading fraction, grain length, and throat to port area. To obtain more candidates, the designers always wish to find a series of suboptimal feasible solutions instead of a single optimal solution.

In this section, taking a given typical slotted-tube grain as the design target, several suboptimal feasible solutions satisfying the burning surface area requirements are found by setting suitable criteria for $C_D$. On one hand, a larger $C_D$ criteria will return grains with a lower loading fraction. On the other hand, a smaller $C_D$ criteria will return grains with a higher loading fraction.

The geometry of the provided slotted-tube grain is shown in Figure 22. The number of the geometry units is 12, and the throat area is $5 \times 10^{-2}$ m. Then, the $\left(\hat{l}_{b,T} C_T\right) \sim \hat{w}_T$ curve is calculated from this grain.

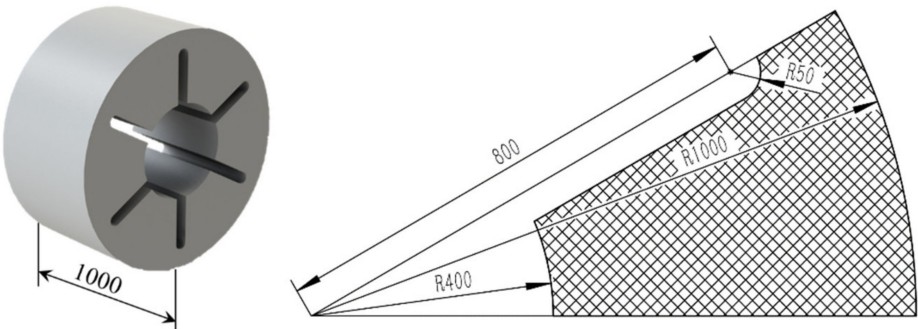

**Figure 22.** Geometry of the given slotted-tube grain.

### 3.2.2. Design Result

The meshing technique and the pre-training of the neural network are not repeated here. By appropriately constraining the range of $C_D$, four suboptimal solutions with different loading fractions can be obtained, as shown in Figure 23. The dimensionless burn perimeter of these schemes is shown in Figure 24.

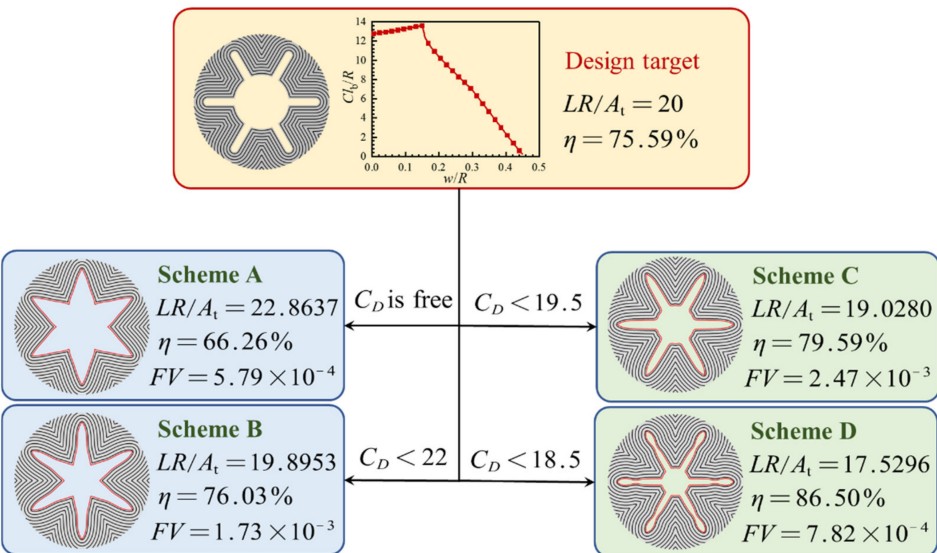

**Figure 23.** Different schemes sharing identical burning surface areas.

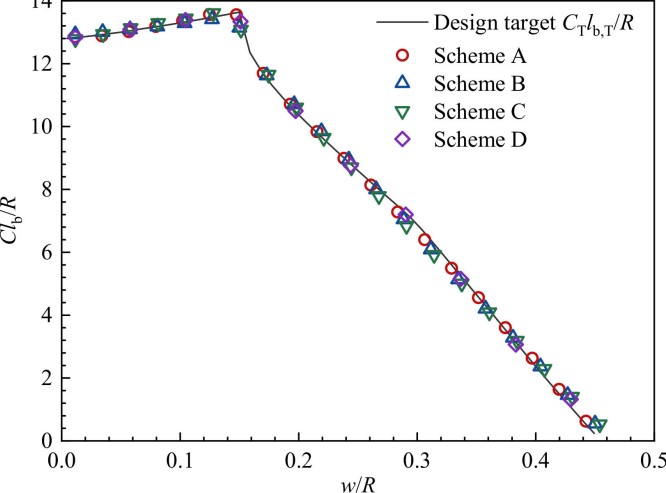

**Figure 24.** Dimensionless burn perimeter of the schemes.

The design result reveals that although the grain shapes of the schemes are quite different, they all share identical burning surface areas or identical performance. The significant difference between them is the loading fraction. The loading fraction of Scheme A is 66.26%, which means the grain length must be long enough to satisfy the performance of the given slotted-tube grain. Contrarily, the loading fraction of Scheme D is 86.5%, which means the grain length can be much shorter. Since the loading fraction of Scheme C and Scheme D are higher than the given one, the method successfully obtains several solutions even better than the original one. Consequently, these schemes could provide designers with more options.

### 3.3. Reverse Design for a Dual-Thrust Goal

3.3.1. Design Target

Dual-thrust motors are widely used in tactical missiles to increase the flight range significantly. The booster stage provides a high but short-term thrust, followed by the cruising stage with low and long-term thrust. Some 2D grain shapes, such as wagon wheel, dendrite, and dog bone [15], might produce dual-thrust internal ballistics. However, there are still questions about whether these grain shapes are the best options. Therefore, it is necessary to design a more suitable grain shape using shape optimization.

The dual-thrust performance curve is given with the aid of a piecewise function shown as Equation (34):

$$p_c(t) = \begin{cases} 10\,\text{MPa}, & \text{if } 0 \leq t \leq 2 \text{ s} \\ (-5t + 20)\,\text{MPa}, & \text{if } 2 \text{ s} < t \leq 3 \text{ s} \\ 5\,\text{MPa}, & \text{if } 3 \text{ s} < t \leq 10 \text{ s} \\ (-5t + 55)\,\text{MPa}, & \text{if } 10 \text{ s} < t \leq 11 \text{ s} \\ 0\,\text{MPa}, & \text{if } t > 11 \text{ s} \end{cases} \tag{34}$$

The parameters of the dual-thrust grain are given in Table 5. According to Equations (6) and (34), and Table 5, the $\left(\hat{l}_{b,T} C_T\right) \sim \hat{w}_T$ curve can be derived from the $p_c \sim t$ curve, as shown in Figure 25.

**Table 5.** Parameters of the dual-thrust grain.

| Symbol | Name | Value | Unit |
|--------|------|-------|------|
| $N$ | Number of units | 12 | — |
| $R$ | Outer radius | 0.5 | m |
| $\rho_P$ | Propellant density | 1700 | kg/m$^3$ |
| $c^*$ | Propellant characteristic velocity | 1400 | m/s |
| $a$ | Burning rate coefficient | $1 \times 10^{-4}$ | m/(s-Pa$^{0.3}$) |
| $n$ | Pressure index | 0.3 | — |

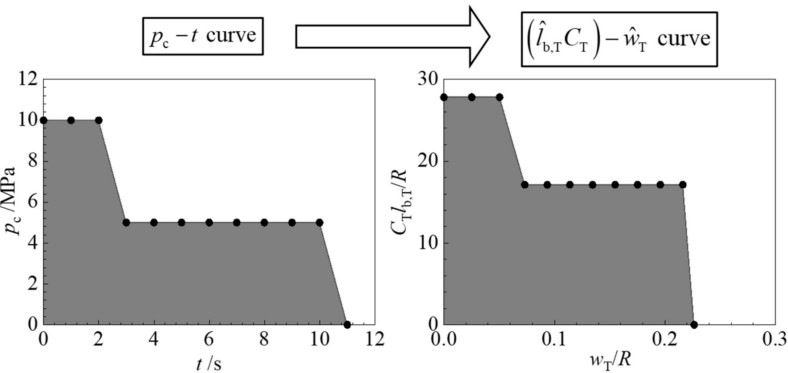

**Figure 25.** Reverse internal ballistic calculations of the dual-thrust grain.

### 3.3.2. Design Result

The evolution process of the dual-thrust grain is illustrated in Figure 26. The figure presents the best individuals in populations during generations. G0 is a typical tube grain whose performance fails to match the given one. After the random population initialization, one individual denoted as G1 among the whole population, appears to match the performance most. Interestingly, the grain shape of G1 seems to be a combination of two different star grains. After 50 generations, the grain shape gradually stabilizes and becomes a star-wagon-wheel combination. After several generations, the results are almost identical, and it is reasonable to believe that a globally optimal solution is obtained. Coincidentally, the design result is similar to a new type of combined dendrite grain proposed by Jiang [1,39] through a traditional approach.

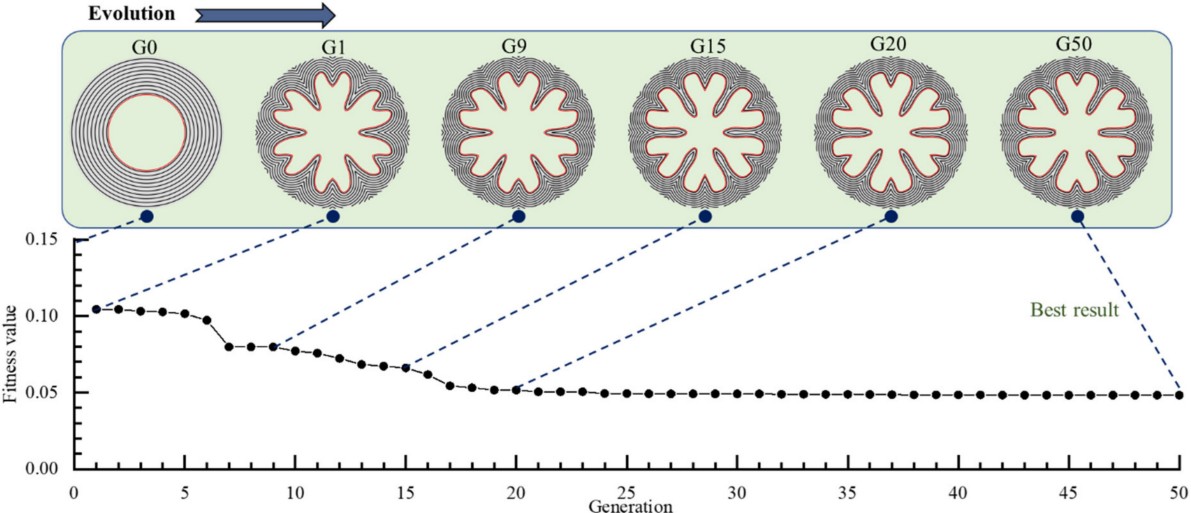

**Figure 26.** Evolution of the designed dual-thrust grain.

Taking G50 as the final design result, as shown in Figure 27, linear regression is used to evaluate the gap between the design result and the target. The result of linear regression is Equation (35):

$$\hat{l}_{b,T} C_T \approx \hat{l}_{b,D} C_D, \ C_D = 31.7668 \tag{35}$$

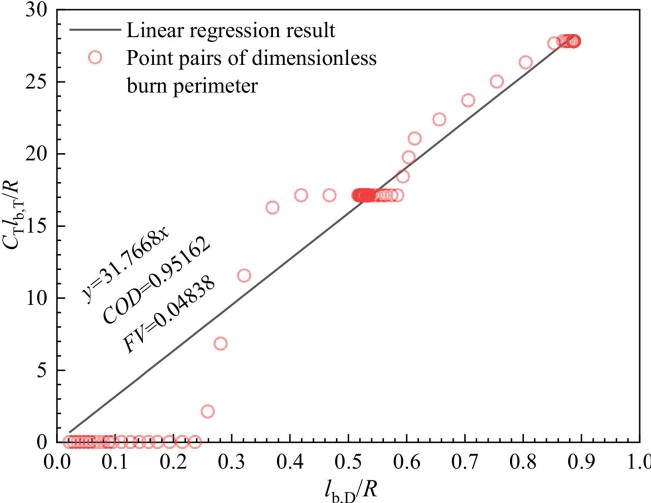

**Figure 27.** Linear regression to evaluate the performance-matching degree of the dual-thrust grain.

The coefficient of determination (COD) is 0.95162, proving the proportional relationship between the two curves. As $R = 0.5$ m, the designed grain must satisfy Equation (36):

$$\frac{L_D}{A_{t,D}} = 63.5336 \text{ m}^{-1} \tag{36}$$

If the outer radius of the grain is fixed to 0.5 m, the length of the grain will strictly correspond to the throat area.

The dimensionless burn perimeter is plotted in Figure 28 as a function of the dimensionless web thickness. Similarly, the combustion chamber pressure is plotted in Figure 29 as a function of time. It can be found that the designed pressure of the booster stage and the cruising stage is stable, which are 10 MPa and 5 MPa, respectively, so the design results have well-matched the expected dual-thrust performance. However, due to the slow burning rate of the residual propellant, the tail-off stage lasts up to 5 s. In order to eliminate the tail-off impulse, it is necessary to replace the residual propellant with a non-flammable filler in further consideration.

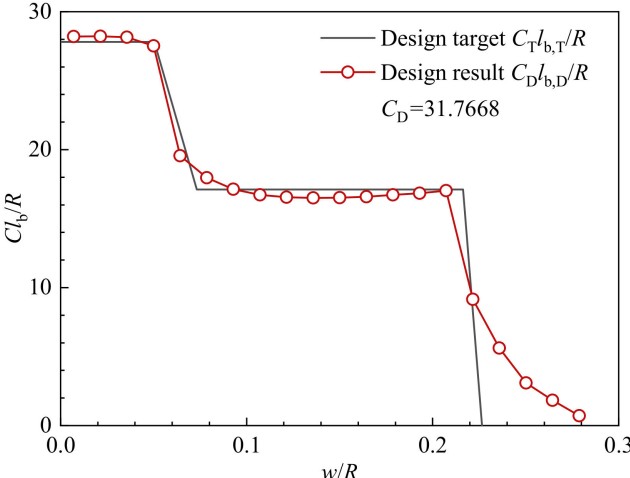

**Figure 28.** Relation between the dimensionless burn perimeter and dimensionless web thickness of the designed grain.

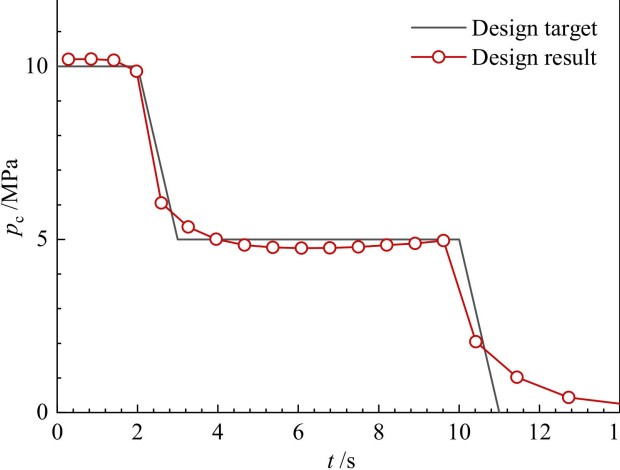

**Figure 29.** Relation between the chamber pressure and the burning time of the designed grain.

It should be noticed that the current results are obtained under the condition of $N = 12$. However, $N$ is not a given integer in practice, so it is worth investigating the design result with different $N$, as shown in Figure 30. The relation between the chamber pressure and the burning time with different $N$ is shown in Figure 31.

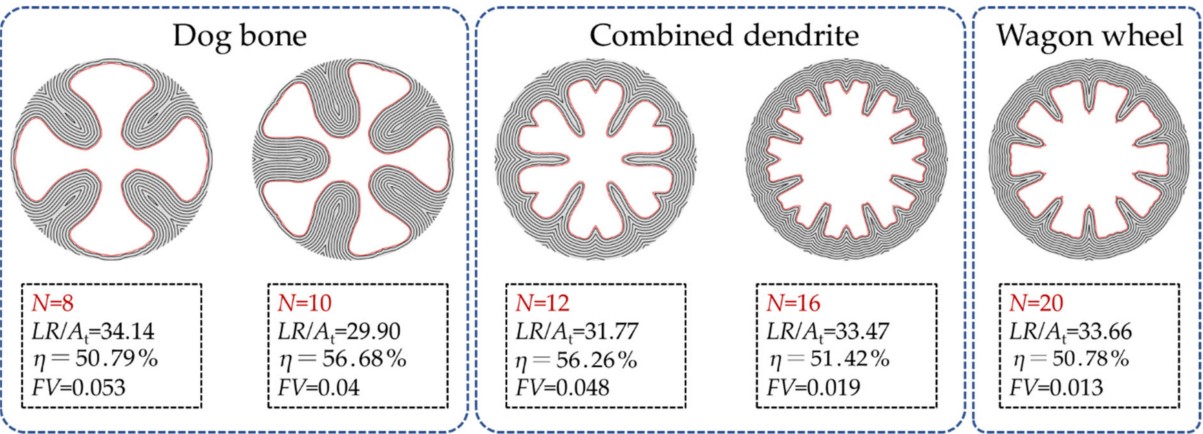

**Figure 30.** Reverse design results with different *N*.

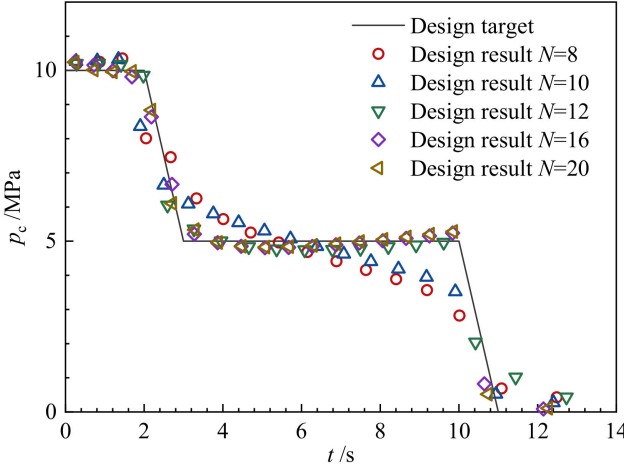

**Figure 31.** Relation between the chamber pressure and the burning time with different *N*.

As can be seen from the above figure, the reverse design result is related to *N*. When *N* is small (8 and 10), the results are similar to the dog bone grain. When *N* is 12 and 16, the results turn into the combined dendrite grain as mentioned before. When *N* is large (greater than 20), the results turn into the wagon wheel grain. As the outer radius of the grain limits the length of the dendritic wheel arm, and if *N* is sufficiently small, the combined dendrite grain will no longer maintain the performance curve, and the design result will turn into a dog bone. It can also be found that with the increase of *N*, the residual propellant becomes smaller, and results with a higher performance-matching degree can be obtained.

In conclusion, the shape optimization method based on the evolutionary neural network can realize the reverse design through an example of a dual-thrust propellant grain. The brand-new grain shape will better match the performance curve and provide designers with numerous innovative ideas.

## 4. Conclusions

The reverse design of solid propellant grain for a performance-matching goal is investigated, and the conclusions are as follows:

1.  The major contribution of this work is to propose a shape optimization method based on the evolutionary neural network to achieve grain reverse design. It does not require any pre-selection of the grain shape and the designers shall be free from defining different kinds of geometric parameters for specific grain configurations.

2. Burn-back analysis method ($\varphi$—PEF) of irregular grain shape on a fixed unstructured mesh is proposed. The method can be directly solved by the steady-state heat conduction modules in commercial finite element platforms.

3. The neural network is designed to determine and manipulate the phase field of the grain. Since the degree of freedom is controllable (much higher), it will form much more grain shapes and gives designers more options.

4. The genetic algorithm framework is established to optimize the hyperparameters in the neural network. The objective function that evaluates the performance-matching degree is the coefficient of determination (COD) derived from the linear regression.

5. The computation results show that the method can precisely match the performance of the given star grain and slotted-tube grain. By adding constraints, it can achieve other performance requirements, such as higher loading fraction.

6. The method can automatically evolve a new dendritic-shaped grain that matches the given dual-thrust pressure-time curve. Consequently, the method has the potential to apply to the conceptual design of innovative grain shapes through the reverse design procedure, which can hardly be predefined and achieved by the traditional approaches.

Future work will focus on extending the method to three-dimensional (3D) grains. Due to the heavy computational burden of 3D problems, the difficulties will lie in improving the time efficiency of the burn-back analysis, introducing a well-designed deep neural network, and adopting a parallel genetic algorithm. In addition, manufacturability, mechanical performance, and new 'pre-set' parameters of grains should also be carefully considered.

**Author Contributions:** Conceptualization, W.L. (Wentao Li) and G.L.; methodology, W.L. (Wentao Li) and W.L. (Wenbo Li); software, W.L. (Wentao Li) and W.L. (Wenbo Li); validation, W.L. (Wentao Li); formal analysis, W.L. (Wentao Li); investigation, W.L. (Wentao Li); resources, W.L. (Wentao Li); data curation, W.L. (Wentao Li); writing—original draft preparation, W.L. (Wentao Li) and W.L. (Wenbo Li); writing—review and editing, G.L. and Y.H.; visualization, W.L. (Wentao Li); supervision, G.L. and Y.H.; project administration, G.L.; funding acquisition, G.L. All authors have read and agreed to the published version of the manuscript.

**Funding:** This research received no direct external funding.

**Institutional Review Board Statement:** Not applicable.

**Informed Consent Statement:** Not applicable.

**Data Availability Statement:** Not applicable.

**Conflicts of Interest:** The authors declare no conflict of interest.

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
