# Peer review of "Reverse Design of Solid Propellant Grain for a Performance-Matching Goal: Shape Optimization via Evolutionary Neural Network"

_aerospace, doi:10.3390/aerospace9100552_

Round 1

Reviewer 1 Report

First of all, I would like to thank you for inviting me to review this interesting research paper. In general, the material brings new and interesting questions related to the use of reverse design of solid propellant grain for a performance-matching goal based on the evolutionary neural network that is proposed to achieve the reverse design of two-dimensional (2D grains).

1.     What is the main question addressed by the research?

The main question adressed by the research is made to the use of the neural network that was introduced to determine the spatial distribution of the propellant to define the grain shape. As described in the text, the hyperparameters of the network are continuously evolved with the aid of the genetic algorithm. Finally, the optimal grain shape that matches the performance goal is most obtained. The method is verified in different scenarios. The result shows that the design can precisely match the given pressure-time curve of star grains and slotted-tube grains.

2. Do you consider the topic original or relevant in the field, and if
so, why?

Yes, i consider the topic original and relevant in the solid propellant grain design mainly by the possibility of the method does not require any pre-selection of the grain shape, and the designers shall be free from defining different kinds of geometric parameters for specific grain configurations. Consequently, the method has the potential to apply in the reconstruction of an actual grain and the conceptual design of innovative grain configurations.

3. What does it add to the subject area compared with other published
material?

The differential of the research is based on the use of evolutionary neural network to achieve the reverse design of two-dimensional (2D) grain. As described in the paper text, given the pressure-time curve, the GA is used to optimize the hyperparameters of the neural network which can fully define the grain shape to minimize the gap between the designed results and the goal. Because the method uses the concept of shape optimization, it does not need to preselect any grain shape, and the designers can be free from defining different kinds of geometric parameters for specific grain shapes.

Another add is the evolutionary process showed in Figure 16. The shape of the grain changes as the GA algorithm proceeds. The figure presents the best individuals in populations during generations. As the result of pre-training, G0 is a typical tube grain. After one generation of reproduction, G1 tends to be a star grain. However, the shape of G5 appears to be highly irregular. After G10, the irregular shape vanishes, and a continuous and smooth star grain is gradually formed

The reverse design for a dual-thrust goal is another important add. As cited in the paper text, dual-thrust motors are widely used in tactical missiles to increase the flight range significantly. The booster stage provides a high but short-term thrust, followed by the cruising stage with low and long-term thrust. Some 2D grain shapes, such as wagon wheel, dendrite, and dog bone [15], might produce dual-thrust internal ballistic. However, there are still questions about whether these grain shapes are the best options. Therefore, it is necessary to design a more suitable grain shape using shape optimization.

4. What specific improvements could the authors consider regarding the
methodology?

The first improvement would be to consider the deformation suffered by the solid propellant grain during burning in the combustion chamber of the solid rocket engine. B Wang, H Peng and YX Chen described the phenomenon in the paper entitled: Effects of deformation on the burning behavior of solid propellants. Abstract In this paper, the burning rate of solid propellants under strains was investigated. In particular, the variations of burning rate at a deformation of 20% or less was measured using a novel apparatus. The results showed that the burning rate increased with increasing strain, but such increase became stable if the strain was increased to a threshold value. In addition, a quadratic functional correlation could be drawn between the burning rate ratio and strain, which was in agreement with the theoretical analysis.

Another improvement is related to changes in ballistic properties when establishing a rotation around the axis of the engine in flight. A.M. Tahsini* and K. Mazaheri addressed this phenomenon in the paper entitled: Swirl Effects on Spinning Solid Propellant Rocket Motor Performance. Abstract: Spining is used in some of solid rocket motors to increase the flight trajectory precision or for stability requirements. The angular acceleration due to the spin increases the burning rate of solid propellant and changes the engine performance by increasing the operating pressure and decreasing the burning time. On the other hand, due to the rocket and its grain rotation, the gases injected from propellant surface to the flow field have an initial angular velocity. The angular velocity of each particle increases as it approaches to the center due to the conservation of angular momentum and a reduction in its radial location, although it vanishes close to the central axis due to viscous effect. These events produce a highly swirling flow in the combustion chamber which changes the radial pressure distribution and discharge coefficient of the nozzle and finally changes the pressure-time history of the engine. Here we have a dual balancing effect between increase of pressure caused by higher burning rate and decrease of pressure due to this swirling effect. The 3D simulation is needed to predict the internal ballistics of these motors accurately and observe simultaneously both of these phenomena. It is achieved here by numerical simulation of Navier_Stokes equations using a finite volume scheme applied on a structured grid. for flow computations, Roe's scheme is used.

An important aspect to be considered is related to the issue of burning instability of the solid propellant grain installed in the combustion chamber because the method does not define previous geometry that can lead to an unstable burning situation depending on the results generated.

5. Are the conclusions consistent with the evidence and arguments
presented and do they address the main question posed?

In general, the conclusions of the work are consistente with the evidence and arguments presented and do they address the main question posed.

In the conclusions, an important consideration is related to future work: Future work will focus on extending the method to three-dimensional (3D) grain. Due to the heavy computational burden of 3D problems, the difficulties will lie in improving the time efficiency of the burn-back analysis, introducing a well-designed deep neural network and adopting a parallel genetic algorithm.

Another important aspect to be considered is related to the fact that the authors could validate the method developed in the work giving a real example using ballistic data from some motor available in the literature. For example, Pengfei REN, Hongbo WANF, Guofeng ZHOU, Jiani LI, Qiang CAI, Jiaquan YU, Ya YUAN carried out similar work validating the method in the paper entitled Solid rocket motor propellant grain burnback simulation based on fast minimum distance function calculation and improved marching tetrahedron method. Abstract: To efficiently compute arbitrary propellant grain evolution of the burning surface with uniform and non-uniform burning rate for solid rocket motor, a unified framework of burning surface regression simulation has been developed based on minimum distance function. In order to speed up the computation of the mini-mum distance between grid nodes of grain and the triangular mesh of burning surface, a fast distance querying method based on the equal size cube voxel structure was employed. An improved marching tetrahedron method based on piecewise linear approximation was carried out on second-order tetrahedral elements, achieved high-efficiency and adequate accuracy of burning surface extraction simultaneously. The cases of star grain, finocyl grain, and non-uniform tube grain were studied to verify the proposed method. The observed result indicates that the grain burnback computation method could realize the accurate simulation on unstructured tetrahedral mesh with a desirable performance on computational time.

6. Are the references appropriate?

Yes, the references used in the work are, for the most part, relevant and current. Therefore, references are relevant.

7. Please include any additional comments on the tables and figures. 

The tables and figures are well dimensioned, in a sufficient quantity to evaluate the stages of the work. 28 figures and 5 tables were counted throughout the text of the work. Some figures, such as 11, could have their size resized due to the large amount of information contained therein. Anyway, there are very few figures that could have their size changed.

Reviewer 2 Report

See attachment. Great work.
